# Antimicrobial, Antioxidant Activities, and HPLC Determination of the Major Components of *Verbena carolina* (Verbenaceae) [note 1]

**DOI:** 10.3390/molecules24101970

**Published:** 2019-05-22

**Authors:** Gonzalo Lara-Issasi, Cecilia Salgado, José Pedraza-Chaverri, Omar N. Medina-Campos, Agustín Morales, Marco A. Águila, Margarita Avilés, Blanca E. Rivero-Cruz, Víctor Navarro, Ramiro Ríos-Gómez, María Isabel Aguilar

**Affiliations:** 1Facultad de Química, Universidad Nacional Autónoma de México, Ciudad Universitaria, México D.F. 04510, Mexico; laraissasi@gmail.com (G.L.-I.); ceci_060192@hotmail.com (C.S.); pedraza@unam.mx (J.P.-C.); omarnoelmedina@gmail.com (O.N.M.-C.); agusmr77@hotmail.com (A.M.); parsel85@hotmail.com (M.A.Á.); catequina@hotmail.com (B.E.R.-C.); 2Jardín Etnobotánico. Centro INAH, Cuernavaca, Morelos 62440, Mexico; avilesmargarita@hotmail.com; 3Laboratorio de Microbiología, Centro de Investigación Biomédica del Sur, Instituto Mexicano del Seguro Social, Xochitepec, Morelos 62790, Mexico; vmnavg@yahoo.com.mx; 4Unidad de Investigación en Sistemática Vegetal y Suelo, FES Zaragoza, Universidad Nacional Autónoma de México, Batalla 5 de mayo y P. Elías calles s/n. Col. Ejército de Oriente, Iztapalapa 09230, Mexico; riosarana44@gmail.com

**Keywords:** *Verbena carolina*, antioxidant capacity, HPLC determination, medicinal plants

## Abstract

*Verbena carolina* L. (Verbenaceae) is used as a decoction in Mexican folk medicine with applications against digestive problems and for dermatological infections. The present work firstly reported HPLC analysis, as well as the free radical scavenging capacity of the extracts and isolated compounds. Antimicrobial analyses of these substances against the bacteria *Staphylococcus aureus*, *Enterococcus faecalis*, *Escherichia coli* and *Salmonella typhi* and the fungi *Candida albicans*, *Trichophyton mentagrophytes* and *T. rubrum* were also tested, as well as the acute oral toxicity in mice of aqueous extracts. Major secondary metabolites in *V. carolina* extracts were isolated by conventional phytochemical methods which consisted of three terpenoids ((1), (3) and (4)) and four phenolic compounds ((2), (4)–(6)). Their contents were determined by HPLC in six different samples from different locations. The results indicated that ursolic acid (1), hispidulin (2), verbenaline (3), hastatoside (4), verbascoside (5), hispidulin 7-*O*-β-d-glucuronopyranoside (6) and pectolinaringenin-7-*O*-α-d-glucuronopyranoside (7) were the main constituents and ranged from 0.17 to 3.37 mg/g of dried plant, with verbascoside being the most abundant and with a significant antioxidant activity in reactive oxygen species (ROS). Hispidulin was the only active compound against *T. mentagrophytes* and *T. rubrum*. The aqueous extract showed no significant toxicity (LD_50_: > 5000 mg/mL). To our knowledge, this is the first comprehensive report of the chemical characterization of *V. carolina* and also of the activity of its constituents towards reactive oxygen species and dermatophytes, and its safety for consumption.

## 1. Introduction

*Verbena carolina* L. (Verbenaceae) is one of the predominant species of the *Verbena* genus that grows in the American continent, and is widely distributed from the southwestern region of the United States to Nicaragua in Central America. In Mexico, this plant is popularly known as ‘verbena’, ‘ajenjo grande’, ‘hierba de San José’, ‘nardo de campo’, ‘Santa María’, ‘poleo negro’ and ‘wahichuri’ (Tarahumara language). It is generally referred to as ‘varvain’ in English. [1,2]. Verbena has a long history of traditional efficacy in Mexico. Most of the plant, except for the roots, is used as a decoction in folk medicine with applications against diarrhea, vomit and dysentery, or as a purgative. Furthermore, the decoction of the aerial parts of *V. carolina* is used to dissolve bladder stones, as a diuretic and to treat wounds, dandruff, allergies and dermatitis [1]. This plant is one of the constituents of a skin care preparation which shows melanogenesis suppression [3]. Favari–Perozzi et al. [2] tested a possible protective effect of verbena extracts in carbon tetrachloride-induced rat liver injury, and Castro et al. [4] determined some of the chemical nutrients, toxic factors, and digestibility of *V. carolina* L. However, the chemical and biological analyses related to the traditional uses and safe prescription of this plant are scarce. Due to the importance of *V. carolina*, it was selected for a program that aims to characterize and include the Mexican medicinal flora in the Mexican Herbal Pharmacopoeia. Accordingly, the purpose of this work is to investigate the major secondary metabolites contained in the extracts of *V. carolina* L., to quantify them by HPLC, and characterize some efficacy parameters such as the free radical scavenging capacity and anti-dermatophyte activity, and also to analyze the safety of its aqueous extracts. To our knowledge, this is reported here for the first time.

## 2. Results and Discussion

### 2.1. Chemical Composition

The isolation and identification of ursolic acid (1), hispidulin (2), verbenalin (3), hastatoside (4), verbascoside (5), hispidulin 7-O-β-d-glucuronopyranoside (6) and pectolinaringenin-7-O-α-d-glucuronopyranoside (7) from the extracts of *V. carolina* was achieved.

From the dichloromethane extract, the chromatographic fraction CDE-3 was submitted to methylation and analyzed by GC-MS. It yielded palmitic, stearic, (Z,Z)-9,12-octadienoic, (Z,Z,Z) 9,12,15-octadecatrienoic and araquidic methyl esters. Fraction CDE-13 eluted with n-hexane–EtOAc (5.5:4.5 *v*/*v*). After methanol recrystallization, this fraction gave ursolic acid ((1), 1.1% yield) which has also been isolated from other Verbena species [5]. The antimicrobial activity of this compound has been reported [6].

A silica gel column chromatography was applied for the acetone extract; fraction CAE-4 provided the flavonoid hispidulin (2), which showed low antimicrobial activity against S. aureus (MIC 100 μg/mL). However, in the assays with the fungi *T. mentagrophytes* and *T. rubrum*, MIC values of 12 μg/mL were obtained. This is important, because one of the traditional uses of the plant is to treat dandruff. Verbenalin (3) was isolated from chromatographic fraction CME-15 as colorless crystals after recrystallization. Fraction CME-20 yielded white crystals that were identified as hastatoside (4). Fraction CME-30 yielded a brownish solid, identified as verbascoside (5).

Compounds 2, 3, 4 and 5 have been isolated from other species of the Verbena genus [5,7,8]. Hispidulin (2) has been isolated from *V. officinalis* and *V. citriodora* [9]; this compound has demonstrated to have potent antioxidant, antifungal, anti-inflammatory, antimutagenic and anticonvulsant activities [10,11,12,13]. It has also showed a strong inhibition of lipid peroxidation in mouse liver homogenates, and has a weak scavenging activity [14]. Hispidulin also exerts anti-osteoporotic and bone resorption inhibiting effects via activation of the AMPK signaling pathway [15]. Hispidulin suppresses the angiogenesis and growth of human pancreatic cancers by targeting the vascular endothelial growth factor receptor [16]. Verbenalin (3) and hastatoside (4) have been reported as sleep-promoting components of *V. officinalis* [17]. In addition, verbenaline (3) showed hepatoprotective activity on experimental liver damage in rodents [18]. Hastatoside (4) was first isolated from *V. hastata* L. and *V. officinalis* L. [19]. Verbascoside (5), isolated for the first time from *Verbascum sinuatum* in 1963 [20,21], is active against *Staphyloccocus aureus* [22], and an inhibitor of protein kinase C [23], it also has anti-inflammatory effects in THP-1 cells [24]. Its structure was confirmed by comparing with literature data [21].

From fraction CME-33, a yellowish precipitate was obtained and identified as hispidulin 7-*O*-β-d-glucuronopyranoside (6). The presence of this compound has also been reported in *V. bonariensis* [25], but in this work, its complete physical and spectroscopic characteristics are reported: m. p. 182–184 °C; [α]25_D_-117.4°; IR: ν_max_ (KBr): 3332, 2922, 1656, 1602, 1509, 1488, 1459, 1351, 1251, 1182, 1066, 1021, 829, 711 cm^−1^; ^1^H-NMR 400 MHz (DMSO-d6): δ: 12.94 (1H, s, C5-OH), 7.75 (2H, d, *J* = 8 Hz, H-6′, H-2′), 6.83 (1H,s, H-8), 6.79 (2H, d, *J* = 12 Hz, H-3′, H-5′), 6.66 (1H, s, H3), 5.12 (1H, d, *J* = 8 Hz, H-1″), 3.78 (3H, s, OCH_3_), 3.70 (1H, d, *J* = 12 Hz, H-5″), 3.34 (2H, m, H-2″, H-3″), 3.26 (1H, dd, *J* = 4, 10 Hz, H-4″); ^13^C-NMR 100 MHz δ: 182.1 (C-4), 172.5 (C-6″), 164.2 (C-2), 162.2 (C-4′), 156.3 (C-7), 152.3 (C-5), 152.0 (C-9), 132.3 (C-6), 128.1 (C-2′), 120.0 (C-1′), 115.8 (C-5′), 105.5 (C-10), 102.0 (C-3), 99.4 (C-1″), 93.9 (C-8), 76.6 (C-3″), 73.8 (C-5″), 72.9 (C-2″), 71.9 (C-4″), 60.2 (-OCH_3_).

From fraction CM-38, a yellowish solid was obtained and identified as pectinolaringenin-7-*O*-α-d-glucuronopyranoside (7). There are reports of a compound very similar to (6), the comantoside B, isolated from *Comanthosphace japonica* (Labiatae) [26] and later by Murata et al. [27]. Contrasting with this, in our work, the flavonoid was identified as the α-glucuronide isomer; its characteristics are as follows: m. p. 201–203 °C; [α]25_D_-34.33°; UV: λ_max_ (MeOH) 271, 330 nm; IR: ν_max_ (KBr): 3332, 2922, 2839, 1656, 1602, 1488, 1351, 1251, 1182, 1066, 1021 cm^−1^; ^1^H-NMR 400 MHz (DMSO-d6): δ: 12.91 (1H, s, H-6″), 8.05 (2H, d, *J* = 12 Hz, H-2′, H-6′), 7.13 (2H, dd, *J* = 8 Hz, H-3′, H-5′), 7.03 (1H, s, H-8), 6.94 (1H, s, H-3), 5.13 (1H, s, H-1″), 3.86 (3H, s, O-CH_3_), 3.78 (3H, s, O-CH_3_), 3.63 (1H, d, *J* = 10, H-5″), 3.23 (2H, m, H-2”, H-3″), 3.22 (H-1, t, *J* = 9.6, H-4″); ^13^C-NMR 100 MHz δ:182.2 (C-4), 171.8 (COOH), 163.6 (C-2), 162.3 (C-4′), 156.5 (C-7), 152.3 (C-9), 132.2 (C-6), 128.2 (C-2′ y C-6′), 122.6 (C-1′), 114.5 (C-3′ y C-5′), 105.5 (C-10), 103.2 (C-3), 99.7 (C-1″), 94.1 (C-8), 76.6 (C-2″), 73.8 (C-3″), 72.8 (C-4″), 71.7 (H-5″), 60.1 (O-CH_3_), 55.4 (O-CH_3_). To the best of our knowledge, this is the first report of hispidulin 7-*O*-β-d-glucuronopyranoside (6) and pectolinaringenin-7-*O*-α-d-glucuronopyranoside (7) in the Verbena species.

Except for hispidulin (3), neither the aqueous extract (CAqE), nor compounds (5), (6) or (7) were active against the tested microorganisms (MIC > 100 μg/mL). The structures of the isolated compounds are shown in Figure 1.

It is worth mentioning that a recent report by Ávila–Reyes et al. [28] only indicates the presence of a flavonoid, a scutellarein glycoside in *V. carolina* foliar tissues. The authors assessed it by comparison of the HPLC retention time and UV absorptions with a standard of this compound, but it was not isolated. Besides this, that report shows no additional information regarding the major chemical composition of this plant.

### 2.2. Phenolic Content of V. carolina Extracts

Total phenolic content in the decoction and in the methanolic extract of *V. carolina* was estimated from the Folin–Ciocalteu method using gallic acid as the standard. The total amount of polyphenols was higher in the aqueous extract than in the methanol one (23.12 and 13.58 gallic acid equivalents/mL). This is an indication of the antioxidant capacity of the extracts, particularly of the decoction, which is employed in its traditional use. These results agree with our phytochemical analysis where the phenolics hispidulin (2), verbascoside (5), hispidulin 7-*O*-β-d-glucuronopyranoside (6) and pectolinaringenin-7-*O*-α-d-glucuronopyranoside (7) were isolated as major compounds.

### 2.3. Validation of the Liquid Chromatography (LC) Method and Analysis of Samples

A reversed phase HPLC system with a dual λ UV detector was used to resolve four compounds in the methanol extract of the aerial parts of the plant. The validation of the method was developed according to the ICH guidelines [29]. Compounds (3), (4), (5) and (6) were detected in a sample of the methanolic extract of *V. carolina* (S1) at retention times (RT) of 9.8, 9.4, 11.6 and 13.6 min, respectively. Four samples of the last solution were then spiked with a known amount of each of the standards, to verify the retention times.

All the calibration curves showed good linearity within the test ranges (R^2^ ≥ 0.999). The linear regression equations for (3)–(6) were y = 50109.4x + 75972.3, y = 30308.0x + 15429.4, y = 38331.8x + 3031.2, and y = 9427.5x + 23078.3, respectively, indicating an excellent correlation between the peak area and concentration.

The LOD and LOQ values (µg/mL) were 2.20 and 2.8 (3), 1.25 and 2.5 (4), 1.40 and 2.8 (5), and 1.25 and 5.0 (6).

The linear regression equations corresponding to the accuracy of the method were y = 1.022x − 0.795, y = 1.004x + 0.050, y = 1.0114x − 0.0412, and y = 1.0095x − 0.8777, for (3)–(6). Percentage recoveries of the compounds are indicated in Table 1. The intra- and inter-day precision RSDs were under 2.0%; the repeatability variation was also less than 2.0%. The results indicate that the method was precise, accurate, and linear for the simultaneous quantification of verbenaline (3), hastatoside (4), verbascoside (5) and hispidulin 7-*O*-β-d-glucuronopyranoside (6) in *V. carolina*.

The developed and validated method was successfully applied for the quantification of (3)–(6) in six samples (S1–S6; Table 2).

In stability tests, the results showed that verbascoside (5) decomposes in the presence of light and heat. Verbenalin (3) is stable below 37 °C for up to seven days.

Compounds (3)–(6) are useful chemical markers for *V. carolina* because they were found in high quantities in all of the analyzed plant samples except for S3 from Hueyapan, Morelos. This sample grows in different environmental conditions (temperature, humidity and altitude). In all cases, verbascoside had the highest concentration. Considering the extent of use of *V. carolina* in Mexican traditional medicine, the content-homogeneity in the samples is of great importance. Furthermore, this study may be useful for the quality control of plant extracts containing compounds (3)–(6). It is also worth noting that (6) was more abundant in a wild sample (S1), than in a cultivated crop (S2) (Table 2 and Figure 2).

### 2.4. Scavenging Activity of V. carolina Extracts

The activity of the *V. carolina* extracts was systematically assessed. Depending on the antioxidant ability of the tested samples, the concentrations used ranged from 0.01 to 0.3 mg/mL. There are reports, e.g., Habu and Ibeth [30] in which 0.4 mg/mL of leave extracts of *Newbouldia laevis* were used to reach about 85% of DPPH (2,2-diphenyl-1-picryl-hydrazil-hydrate) radical scavenging. Also, Okoh et al. [31] employed up to 0.5 mg/mL of seed and shell essential oils obtained from the climbing vine *Abrus precatorius* L. in ABTS (2,2′-Azino-bis(3-ethylbenzothiazoline-6- sulfonic acid) and DPPH radical assays. The results of three in vitro assays (DPPH, ABTS, FRAP (Ferric reducing ability of plasma)) measuring the antioxidant activities are given in Table 3. The TEAC (Trolox equivalent antioxidant capacity) values of the ABTS and DPPH assays showed that the methanol extract (CME) stabilized the radicals better than the aqueous extract (CAqE) (10.10 vs. 7.10 and 14.20 over 7.50 respectively). Ascorbic acid was used as standard compound in both assays. The results of the FRAP assay are expressed as μmol TEAC per mg of extract; again, CME better stabilizes radicals than CAqE (11.74 vs. 5.49). Quercetin was used as standard in this assay. In all cases, the antioxidant capacity was lower than that of the standards (Table 3).

### 2.5. Scavenging Activity of V. carolina Bioactive Compounds: Comparison Against Extracts

Based on the EC_50_ values of the tested samples, additional tests were performed with specific ROS as ROO^•^, O_2_^•−^, H_2_O_2_, OH^•^, ^1^O_2_, HClO, and ONOO^−^. The analysis included CAqE, CME, verbascoside (5), verbenalin (3), hastatoside (4) and hispidulin 7-*O*-β-d-glucuronopyranoside (glucuronide 7OβGH, (6)).

The methanol extract (CME) was more active than aqueous one (CAqE) in most cases. Regarding the isolated compounds, verbascoside (5) presented the greater quenching, which may be due to the catechol residues in both of its aromatic rings.

Verbenalin (3) was active against ROO^•^, O_2_^•−^, OH^•^ and ONOO^−^ and hastatoside (4) against ROO^•^, OH^•^ and ONOO^−^. These two iridoid compounds were less active than glucuronide 7OβGH (6) and verbascoside (5). Additionally, (3), (5) and (6) showed higher antioxidant abilities than Tiron (Figure 3).

Regarding the HClO scavenging, verbascoside (5) showed the highest capacity, followed by CME, CAqE and glucuronide 7OβGH (6). Some of the popular uses of the plant such as hepatic protection and wound healing could be attributed to the antioxidant properties showed.

### 2.6. Antimicrobial Activity and Acute Toxicity

The dilution method used in this work is frequently recommended for establishing the relative antimicrobial potency of complex plant extracts and their antimicrobial spectrum [32]. There is not a uniform criterion to compare the antimicrobial activity of extracts with that of reference antibiotics; literature data indicate that MIC values between 2.5 and 15 mg/mL [33] may lead to strong antimicrobial compounds [34,35].

Then tested extracts showed antimicrobial activity against the dermatophytes (*T. mentagrophytes* and *T. rubrum*), in addition, the methanol extract (CME) was active against *C. albicans* (MIC values of 0.7 mg/mL). All the extracts (CAqE, CDE and CME) were only slightly active against *Enterococcus faecalis* and *Staphylococcus aureus* (MIC 1.5 mg/mL), the methanol extract (CME) also showed low activity against *Salmonella typhi* (MIC of 1.5 mg/mL). No antibacterial activity against *Escherichia coli* was observed in any of the extracts or isolated compounds.

For the acute toxicity test, doses of 10, 100 and 1000 mg/kg of CAqE in the first phase, and 1600, 2900 and 5000 mg/kg in the second phase were employed; an LD_50_ value higher than 5000 mg/kg was calculated using the geometric mean of the doses for which none of the animals died (0/3 deaths were found), and none of the organs analyzed after the experiment showed physical anomalies. Hence, CAqE of *V. carolina* could be regarded as non-toxic [36].

## 3. Materials and Methods

### 3.1. Chemicals and Materials

Sodium pyruvate, dimethyl thiourea (DMTU), ascorbic acid, 2,2′-azinobis-(3-ethylbenzothiazoline-6-sulfonic acid) (ABTS), 2,2-diphenyl-1-picrylhydrazyl (DPPH), dimethylsulfoxide (DMSO), nitroblue tetrazolium (NBT), potassium nitrite (KNO_2_), manganese dioxide (MnO_2_), diethylene triamine pentaacetic acid (DTPA), DL-penicillamine, potassium persulfate (K_2_SO_4_), sodium carbonate (Na_2_CO_3_), 4-aminobenzoic-acid, gallic acid, 4-(2-hydroxyethyl)-1-piperazineethanesulfonic acid, 6-hydroxy-2,5,7,8-tetramethylchroman-2-carboxylic acid, *N*-acetyl-3,7-dihydroxyphenoxazine (Amplex Red), 2,2′-azobis(2-amidinopropane) dihydrochloride (AAPH), iron chloride hexahydrate, dichloro-dihydro-fluorescein diacetate (DCDHF-DA), 1,3-diphenyl isobenzofurane (DPBF), fluorescein, phenazine methosulfate (PMS), β-nicotinamide adenine dinucleotide (NADH), horseradish peroxidase, Folin–Ciocalteu reagent, 4-(2-hydroxyethyl)-1-piperazineethanesulfonic acid (HEPES), 4,5-dihydroxy-1,3-benzenedisulfonic acid disodium salt monohydrate (Tiron), 2,4,6-Tris(2-pyridyl)-s-triazine (TPTZ) and terephthalic acid (TA) were purchased from Sigma Aldrich (St. Louis, MO, USA), and gentamicin and miconazole were purchased from Sigma Aldrich (St. Louis, MO, USA). Sodium hydroxide was obtained from Meyer (Mexico City, Mexico). Absolute ethanol, hydrogen peroxide (H_2_O_2_), methanol (MeOH), ethylenediamine-tetraacetic acid disodium salt (EDTA), sodium hypochlorite (NaOCl) and sodium nitrite (NaNO_2_) were purchased from JT Baker (Mexico City, Mexico). All other chemicals were reagent grade and commercially available. Acetonitrile (LiChrosolv ^®^) was HPLC grade and deionized water was obtained from a MilliQ System. Pure verbenalin (3), hastatoside (4), verbascoside (5) and hispidulin 7-*O*-β-d-glucuronopyranoside (6) (>98% determined by HPLC) were isolated from *Verbena carolina* and used for the calibration. Their identities were confirmed based on spectral data (IR, UV, NMR and MS) [5,7,8,9].

### 3.2. Plant Material

Aerial parts of *Verbena carolina* were either collected or locally purchased as follows: Samples 1 (S1) and 2 (S2) from Mercado de Sonora (wild and cultivated, respectively, 2013), sample 3 (S3) from Hueyapan, Morelos, Mexico (April 2005), samples 4 (S4), 5 (S5) and 6 (S6) from State of Mexico (2013, S4: 19° 5′ 17.64″ N, 99° 33′ 40.48″ O, at 2600 masl, S5: 19° 5′ 26.85″ N, 99° 34′ 41.41″ O, at 2800 masl, and S6: 19° 6′ 51.45″ N, 99° 35′ 9.15″ O, at 2600 masl). The material was identified by M. Avilés and M. Fuentes and a voucher specimen (INAHM-2018) was deposited in the Herbarium of the Instituto Nacional de Antropología e Historia Morelos (INAHM) in the Medicinal Botanical Garden in the city of Cuernavaca, Morelos, Mexico.

### 3.3. Preparation of the Extracts and Isolation of Compounds

The air-dried aerial parts of *V. carolina* L. (1.3 kg, S1) were ground into powder and successively macerated at room temperature with n-hexane, dichloromethane, acetone and methanol (48 h with 4 L for each solvent) and filtered. This was repeated thrice and each extract was combined and evaporated under vacuum to yield the corresponding hexane (CHE, 17.7 g), dichloromethane (CDE, 174.9 g), acetone (CAE, 90 g) and methanol (CME, 131.0 g) syrupy residues. Another batch of *V. carolina* (50 g, S1) was extracted by decoction with water and concentrated in vacuo to give 11 g of a concentrated residue (CAqE). CAqE, CDE and CME showed MIC values ranging between 0.7 and 3.0 mg/mL in the antimicrobial assay, thus, a bioguided fractionation was done based on the antimicrobial properties, particularly the activity against dermatophytes, using the agar dilution method. CDE (174.8 g) was subjected to column chromatography over silica gel 60 (1.0 kg). A sample of fraction CDE-3 (20 mg), which eluted with hexane-ethyl acetate (EtOAc) 90:10, was methylated with 2 mL of 5% KOH in methanol at 80 °C for 1 h in a sealed tube. Then, 100 µL of boron trifluoride (BF_3_) were added and the mixture heated for an additional hour at 80 °C. After cooling at room temperature, 4 mL of water were added and the mixture partitioned with toluene-hexane (8:2 *v*/*v*). The residue of the evaporation of the organic layer was analyzed by gas chromatography–mass spectrometry (GC-MS); a mixture of fatty acids was identified. From fraction CDE-13 (1.6 g, hexane-EtOAc 55:45 *v*/*v*) a white powder precipitated; it was recrystallized with MeOH and identified as ursolic acid (1) by spectroscopic techniques [6].

In a separate experiment, CAE (90 g) was dissolved in distilled water (1 L), filtered and successively partitioned with EtOAc and n-butanol. This last extract was evaporated (13.9 g) and subjected to column chromatography (SiO_2_, 231 g) with a gradient of CH_2_Cl_2_/MeOH (10:0–0:10 *v*/*v*). Fractions of 100 mL each were collected and pooled based on the system of solvents used to yield ten main fractions (CAE-1–CAE-10). From fraction CAE-4 (680 mg), a yellow powder spontaneously precipitated, which upon EtOH recrystallization gave 200 mg of hispidulin (2).

In a third experiment, 50 g of CME were subjected to column chromatography (SiO_2_, 200 g), eluting with gradients of CH_2_Cl_2_/MeOH (10:0–0:10 *v*/*v*) to yield 40 primary fractions (F1–F40) of 100 mL each. Verbenaline (3) was obtained from fraction CME-15 (9.5:5) as a white solid (700 mg) and was recrystallized with n-hexane-EtOAc; hastatoside (4) was also obtained as a white solid (50 mg) from fraction CME-20 (CH_2_Cl_2_/MeOH 9:1 *v*/*v*); verbascoside (5) was obtained from fraction CME-30 (CH_2_Cl_2_/MeOH 8:1 *v*/*v*) as a brown solid (1 g). Compounds (4) and (5) were purified with a secondary SiO_2_ column chromatography (EtOAc 100%). Hispidulin 7-*O*-β-d-glucuronopyranoside (7OβGH, (6)) and pectinolaringenin-7-*O*-α-d-glucuronopyranoside (7) were respectively isolated from fractions CME-33 (20 mg) and 38 (40 mg) (CH_2_Cl_2_-MeOH, 8.0:2.0 *v*/*v*), as yellowish precipitates. The chemical identities of all the isolated compounds were confirmed by spectroscopic techniques.

### 3.4. Total Phenolics Quantitation

Total polyphenols were determined in the aqueous (CAqE) and methanolic (CME) extracts by the Folin–Ciocalteu method. Each extract solution (20 µL) containing 1 mg of plant extract was mixed with 160 µL of distilled water and 20 µL of the Folin–Ciocalteu reagent. The mixture was incubated for 8 min at room temperature in the dark, and 10 µL of sodium carbonate (20%) were added. The mixture was incubated for 1 h at room temperature and absorbance at 760 nm was measured. Total polyphenols were expressed as mg of gallic acid equivalents/g of extract after interpolating in a calibration curve.

### 3.5. Equipment and Chromatographic Conditions

IR spectra were obtained using KBr disks or films on a Perkin–Elmer FT 1605 spectrophotometer. NMR spectra including COrrelation SpectroscopY (COSY), Nuclear Overhauser Effect SpectroscopY (NOESY), Heteronuclear Multiple Bond Correlation (HMBC) and Heteronuclear Single Quantum Correlation (HSQC) experiments were acquired on a Varian Unity INOVA at 300 or 400 MHz (^1^H) and 75 or 100 MHz (^13^C). Electron impact mass spectrometry (EI-MS) was recorded on a JEOL SX 102A mass spectrometer and optical rotations were determined on a Perkin–Elmer Model 241 polarimeter. For open column chromatography, SiO_2_ 60 (70–230 mesh, Merck, Germany) was used and silica gel 60 F254 (Merck) for TLC.

### 3.6. Preparation of Stock and Working Solutions for HPLC

Stock solutions of compounds (3)–(6) were prepared in methanol at 210, 120, 270 and 240 µg/mL, respectively. The calibration curves for each standard were made by stepwise dilution of the stock solutions to obtain seven different concentrations in the range of (µg/mL): 8.8 to 210 (3), 5 to 120 (4), 11 to 270 (5) and 10 to 240 (6).

Air-dried parts of *V. carolina* were ground and sifted through a 1.4 mm sieve. The obtained powder (50 mg) was mixed with 3 mL of methanol in a conical flask and sonicated for 15 min. The extract was centrifuged (15 min, 3000× *g*) and the supernatant diluted to 10 mL. Then, it was filtered through a 0.45 μm polyvinylidene difluoride (PVDF) membrane syringe filter prior to injection into the HPLC system.

### 3.7. HPLC Analytical Method

The HPLC system (Waters Corp., Milford MA, USA) consisted of a 515 pump, an AS-2055PLUS automated injector (Jasco), a 680 automated gradient controller, a 2487 two-channel UV/visible (VIS) detector, and a computerized data station equipped with the Waters Millennium software. All the analyses were performed using a C18 stationary phase in a PrincetonSPHER 100 Å column (150 × 4.6 mm I.D.; 3 μm particle size) operating at 30 °C. The mobile phase consisted of 0.1% acetic acid in water (A) and acetonitrile (B) with the following gradient (*v*/*v*): From 96:4 to 80:20 in 8 min; then to 60:40 in 6 min, to 35:65 in 2 min and finally to 0:100 in 4 min at 0.7 mL/min. Compounds (3)–(5) were monitored at 240 and 330 nm (volume of injection 20 μL). Each run was followed by a 5 min wash with 100% acetonitrile and an equilibration period of 15 min. The compounds were identified based on the retention time and by spiking with standards under the same conditions.

The method was validated according to the International Council for Harmonisation (ICH) guidelines. The following characteristics were evaluated: Selectivity, accuracy, precision, linearity, and limits of detection (LOD) and quantitation (LOQ). The selectivity was assessed based on the peak purity and resolution from the nearest eluting peaks.

The calibration curves were prepared in the concentration range expected for each compound as described in Section 3.6. The correlation coefficient was obtained from the peak area vs. concentration plots using the Statgraphics Centurion XV software (Statpoints Technologies, Warrenton, VA, USA).

The precision was determined by evaluating the repeatability and intermediate precision using the standard addition method. For the former a set of six replicate analyses of the same solution containing 70, 40, 90 or 80 μg/mL of (3)–(6), respectively, was evaluated in a single day. The intermediate precision was estimated from a set of six-fold analyses of the same (3)–(6) solution on different days and by two analysts. The standard deviation (SD) and coefficient of variation (RSD) were calculated for each analysis

Recovery experiments were performed to assess the accuracy of the method. Three concentrations (low, medium, and high) were selected over the linear range. All the analyses were performed by spiking (3)–(6) to the samples, and then evaluating the recovery. The limits of detection (LOD) and quantitation (LOQ) were respectively calculated as 3.3 and 10 times the SD.

The validated analytical method was used for the simultaneous determination of (3)–(6) in six batches sampled from several provinces (S1–S6). The six dry samples (S1–S6) of *V. carolina* were powdered, extracted, and analyzed as described in Section 3.6.

The stability of the samples was evaluated under different conditions: Room temperature, darkness, refrigeration, refrigeration and darkness, and 37 °C in darkness. Each one was analyzed at 24 and 72 h, and 8 days.

### 3.8. Radical Scavenging Tests

To evaluate the DPPH scavenging ability, the method described by Cheng et al. [37] was used employing six concentrations of CAqE and CME ranging from 0.01 to 0.3 mg/mL and 100 μL of a solution of methanol and DPPH (0.208 mM). The control sample contained distilled water. After incubation for 20 min at room temperature and in the dark the absorbance was recorded at 515 nm. The scavenging capacity is given as the percentage (%) of DPPH scavenged, calculated as [(optical density of control—optical density of compound)/(optical density of control) × 100]. EC_50_ values were obtained correlating the DPPH radical scavenging percent and the concentration of samples. The ability of *V. carolina* extracts to scavenge DPPH was compared with that of ascorbic acid.

ABTS radical cation was generated by the incubation of 7 mM ABTS with 2.5 mM potassium persulfate in the dark at room temperature for 12–16 h [38]. The ABTS^•+^ solution was diluted to an absorbance of 0.70 at 734 nm in methanol. Then, 20 µL of CAqE or CME extracts ranging from 0.01 to 0.6 mg/mL were added to 180 µL of the ABTS^•+^ solution and the absorbance was read after six min. Ascorbic acid was used as a reference for the scavenging activity.

For the ferric reducing antioxidant power (FRAP) assay, the method by Benzie and Strain [39] was employed. The FRAP solution was prepared mixing a 300 mM acetate buffer solution (adjusted to pH 3.6) with 20 mM ferric chloride hexahydrate dissolved in distilled water and 10 mM TPTZ dissolved in 40 mM HCl. Afterwards, 20 µL of five concentrations of either CAqE or CME from 0.01 to 1.0 mg/mL were added to 180 µL of the FRAP solution. After incubation for 30 min at room temperature in darkness, the absorbance was recorded at 595 nm. Trolox and ascorbic acid were used to obtain the calibration curves and quercetin as the standard compound.

### 3.9. Reactive Oxygen Species (ROS)-Scavenging Measurements

The ROS-scavenging capacity of the aqueous (CAqE) and methanol (CME) extracts were tested in seven in vitro assays for the following species: Peroxyl radical (ROO^•^), superoxide radical (O_2_^•−^), hydrogen peroxide (H_2_O_2_), hydroxyl radical (^•^OH), singlet oxygen (^1^O_2_), hypochlorous acid (HClO) and peroxynitrite (ONOO^−^). All the measurements were carried out in 96 wells plates and based on previously reported methods with minor modifications and they were assayed in triplicate. Briefly, for the peroxyl radical scavenging activity, the method was based on the peroxyl-radical mediated damage of fluorescein [40,41]. Samples were tested at nine concentrations from 0.002 to 0.5 mg/mL in distilled water. Readings were recorded at 535 nm (emission) and 485 nm (excitation). Trolox was used for the calibration curves.

For the superoxide scavenging analysis, a colorimetric assay was performed as follows: 10 μL of DMSO were added to 1.14 mL of 20 mM HEPES buffer, pH 7.2 containing 5 μM PMS, 50 μM NBT and samples (CAqE and CME) at six concentrations from 0.5 to 5 mg/mL. To the mixture, 50 μL of 2.5 mM NADH solution were added to initiate the generation of the superoxide anion. After the addition, the optical density at 560 nm was recorded [42]. Tiron was used as the standard for the calibration curves.

The hydrogen peroxide scavenging fluorimetric assay relies on the reaction between Amplex Red and H_2_O_2_, which is catalyzed by horseradish peroxidase [43]. The aqueous solutions of CAqE and CME at six concentrations (0.5 to 5 mg/mL) were mixed with 10 µM H_2_O_2_. After 30 min, the H_2_O_2_ concentration was estimated by measuring the fluorescence using 530/25 excitation and a 590/35 emission filter. Sodium pyruvate was used as the standard.

The hydroxyl radical scavenging capacity was evaluated according to [43]. The OH^•^ was generated by the Fe^3+^–EDTA–H_2_O_2_ reaction. Terephtalic acid, which is not fluorescent, was used to assess the generation of the radical since it reacts with OH^•^ to form the fluorescent 2-hydroxy-TA. Briefly, 180 µL of the following reaction mixture: 0.2 mM ascorbic acid, 0.2 mM FeCl_3_, 0.208 mM EDTA, and 1 mM H_2_O_2_, 1.4 mM TA in 20 mM phosphate buffer (pH 7.4) were mixed with 20 μL of different concentrations of CAqE and CME, ranging from 0.5 to 5 mg/mL. The increase in fluorescence emission at 432 nm was measured for 30 min after a 326 nm excitation. DMTU was used as the standard.

To measure the singlet oxygen scavenging activity, a solution of ethanol, 1 mM H_2_O_2_, 3 µM NaOCl, and 1 µM DPBF was mixed with five aqueous solutions of CAqE and CME (0.3125–5 mg/mL). The plates were incubated for 10 min at 25 °C in the dark, then 150 μL of ethanol were added to each well and the fluorescence emission at 455 nm was measured after excitation at 410 nm [43]. Data were collected every minute for 10 min. Lipoic acid was used as the standard.

For the hypoclorous acid scavenging assay, 0.5 to 5 mg/mL solutions of CAqE or CME were mixed with 30 μL of 0.05 mM HClO and with 30 μL of 0.1 mM p-aminobenzoic acid. The fluorescence was registered for 15 min using filters for a 280 nm excitation and 340 nm emission. Ascorbic acid was used as the standard.

For the peroxynitrite scavenging assay, CAqE or CME were tested at concentrations from 0.1 to 5 mg/mL in a solution containing 33 µM DTPA, 100 µM DCDHF-DA, and 20 µM ONOO^−^ in 0.1 M sodium phosphate buffer pH 7.4. Fluorescence was measured at 528 nm after excitation and at 485 nm emission every five min for 30 min [41]. Penicillamine was used as standard. Peroxynitrite was synthesized as follows: 5 mL of an acidic solution of 0.7 M H_2_O_2_ (in 0.6 M HCl) were mixed with 5 mL of 0.6 M KNO_2_ on an ice bath for 1 s before the reaction was quenched with 5 mL of ice-cold 1.2 M NaOH. Residual H_2_O_2_ was removed using a column of granular MnO_2_ prewashed with 1.2 M NaOH and the filtered solution was then left overnight at −20 °C. The resulting yellow liquid layer on the top of the frozen mixture was collected for the assays. The concentration of ONOO^−^ was determined before each experiment at 302 nm considering a molar extinction coefficient of 1670 M^−1^ cm^−1^.

### 3.10. Antibacterial and Antifungal Activity

For the antibacterial and antifungal assays, the following strains, purchased from American Type Culture Collection (ATCC, Manassas, VA, USA), were used: *Staphylococcus aureus* ATCC 6358; *Enterococcus faecalis* ATCC 10231; *Escherichia coli* ATCC 8937; *Salmonella typhi* ATCC 06539; the filamentous fungi *Trichophyton mentagrophytes* ATCC 28185 and *Trichophyton rubrum* ATCC 28188 and *Candida albicans* ATCC 10231. The antifungal assay was done in Petri dishes (Falcon) [44,45]. The antibacterial assay was carried out via the agar dilution method [46]. Both the antibacterial and antifungal activities were measured by determining the MIC of CAqE and compounds (1)–(7).

For the antibacterial assay, samples were dissolved in DMSO (2% *v*/*v*) at 10 mg/mL. Then, the solutions were diluted to reach final concentrations of 1.5 to 400 µg/mL. The inoculum for each organism was prepared from cultures containing 108 colony-forming units (CFU)/mL. The diluted (1:20) inoculum was applied as a spot using a calibrated loop that delivered 0.002 mL, resulting in a circular spot of 5–8 mm diameter with 104 CFU. The plates were incubated for 24 h at 37 °C. Gentamicin (2.5–120 µg/mL) (Sigma) was used as the standard. Observations were performed by duplication. Results are expressed as the lowest concentration of plant extract that completely suppresses colony growth on the agar.

For the antifungal assay, the samples and reference compounds, Nystatin (Merck) and Miconazole (Sigma), were used in two-fold serial dilutions yielding concentrations in the range from 1.5 to 400 µg/mL for extracts and from 1.0 to 128 µg/mL for pure compounds; 8 µg/mL and 4 µg/mL for Nystatin and Miconazole, respectively. Final concentrations of DMSO in the test were less than 2% (*v*/*v*). A final inoculum of 105 cell/mL for *Candida albicans* and 106 spore/mL for the filamentous fungi was spouted on top of the solidified agar with a loop calibrated to deliver 0.005 mL. Experiments were carried out during duplication with incubation at 29 °C. The fungal growth was assessed in control plates prepared without any test samples after 24, 48 and 72 h, depending on the incubation period required for a visible growth: 24 h for *Candida albicans*, and 72 h for the dermatophytes.

### 3.11. Animals

ICR male mice (25–30 g) obtained from Centro UNAM-Harlan (Harlan México, S.A. de C.V.) were used for the toxicity studies. Procedures involving animals and their care were conducted according to protocols approved by the Universidad Nacional Autónoma de México, Facultad de Química Animal Care Committee under the Mexican Official Norm for Animal Care and Handling (NOM-062-ZOO-1999) and in compliance with the international rules on care and use of laboratory animals.

### 3.12. Acute Toxicity Study in Mice

Food was withheld 12 h before the experiment, but animals had water ad libitum. Two groups of three mice each were orally fed with the *V. carolina* decoction (CAqE) at doses of 10, 100 and 1000 mg/kg in a first phase, and 0, 1600, 2900 and 5000 mg/kg in a second phase. Animals were observed daily for 14 days for mortality, toxic effects and/or changes in behavioral pattern, according to Lorke’s method [36,47]. At the end of the experiments the animals were euthanized in a CO_2_ chamber. A post mortem analysis of all animals was practiced evaluating the physical characteristics of the main vital organs (heart, lungs, liver, stomach, bowels and kidneys).

### 3.13. Statistical Analysis

Data are expressed as mean ± SEM (standard error of the mean) and were compared against the blank tube without *V. carolina* extracts. The scavenging capacity was expressed as the 50% effective concentration (EC_50_) value, which denotes the concentration of *V. carolina* extracts, required for a 50% reduction in the oxidizing effect relative to the blank tube.

## 4. Conclusions

The beneficial effect of CAqE for dermatological conditions may be justified by the results obtained in the antifungal and antioxidant assays. With respect to antimicrobial activity, hispidulin (2) was active against the fungi *T. mentagrophites*, *T. rubrum* and *C. albicans*. Regarding the scavenging effect, in general, individual compounds were more effective than extracts. The results also support the widespread use of this plant in traditional medicine.

A simple and reliable RP-HPLC analytical method was developed and fully validated for the quantification of verbenaline (3), hastatoside (4), verbascoside (5) and hispidulin 7OβGH (6) in *Verbena carolina*. The method was used to establish seasonal and geographical variations of the markers in *V. carolina* from several regions of Mexico. These marker compounds from the plant should be useful for quality control procedures which are necessary for medicinal plants authentication protocols, such as monographs.

## Figures and Tables

**Figure 1 molecules-24-01970-f001:**
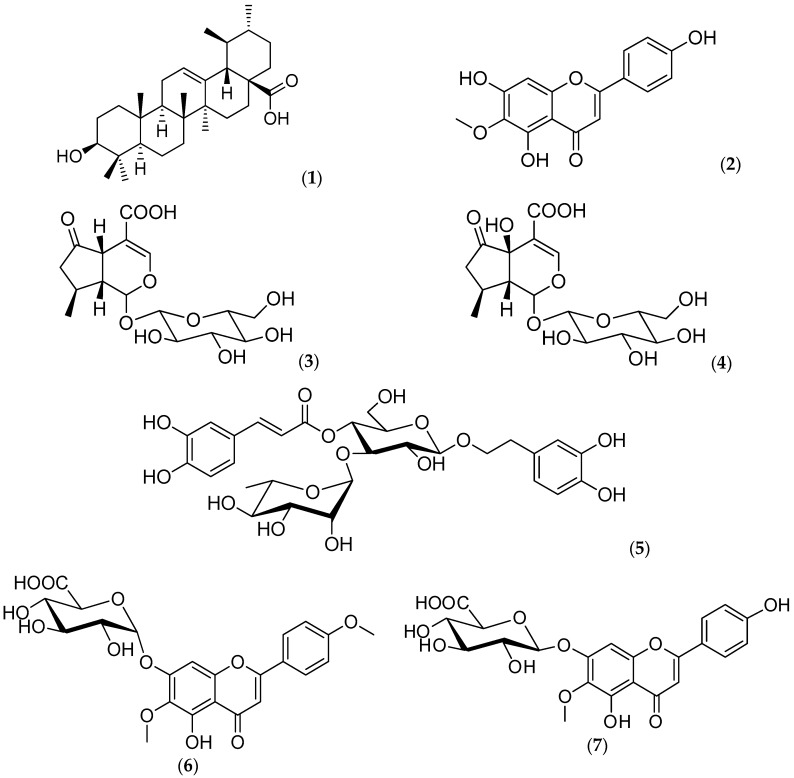
Chemical structures of compounds isolated from *V. carolina*. Ursolic acid (1), hispidulin (2), verbenaline (3), hastatoside (4), verbascoside (5), hispidulin 7-*O*-β-d-glucuronopyranoside (7OβGH, (6)), pectinolaringenin-7-*O*-α-d-glucuronopyranoside (7).

**Figure 2 molecules-24-01970-f002:**
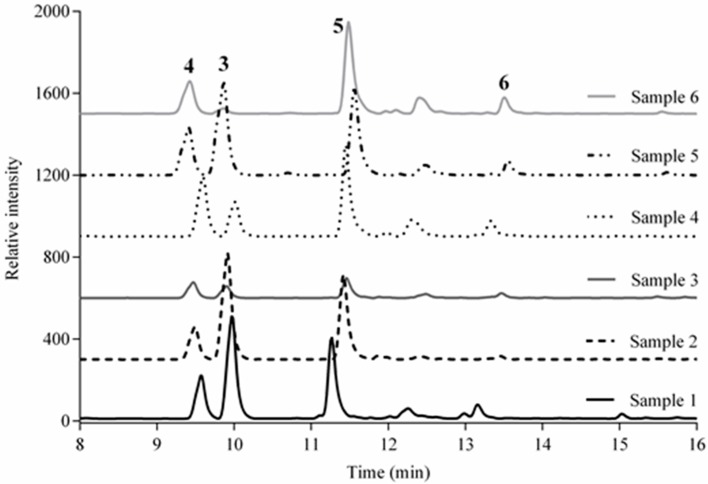
Chromatograms of samples (1)–(6).

**Figure 3 molecules-24-01970-f003:**
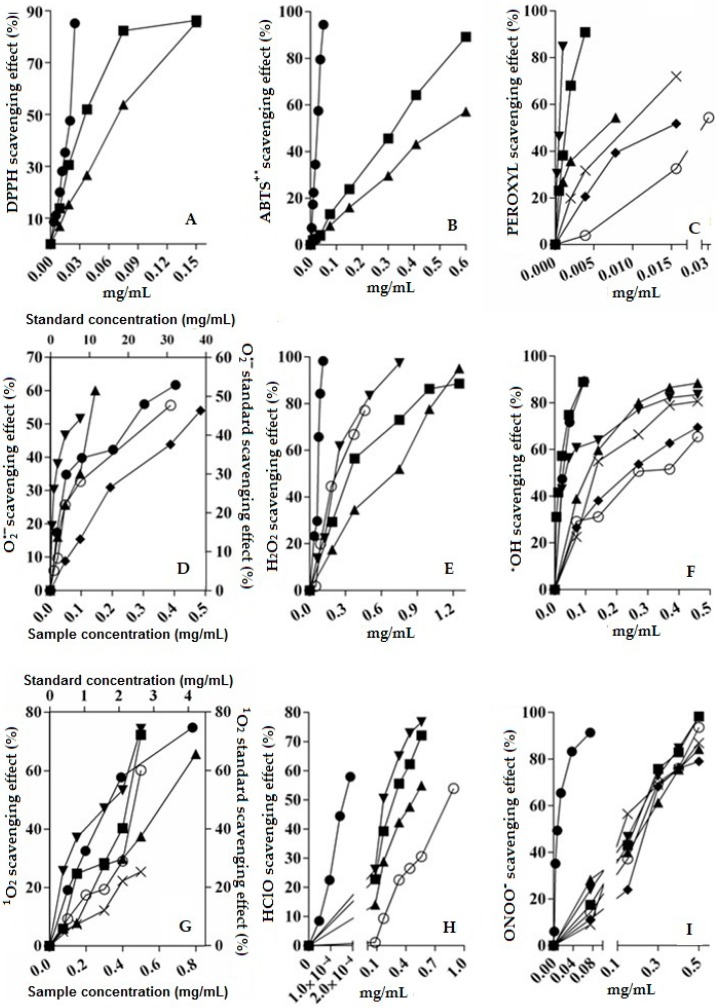
Antioxidant properties of the extracts and the marker compounds (3)–(6) of *V. carolina*. DPPH (panel **A**), ABTS (panel **B**), ROO^●^ (panel **C**), O_2_^●−^ (panel **D**), H_2_O_2_ (panel **E**), OH^●^ (panel **F**), ^1^O_2_ (panel **G**), HClO (panel **H**), ONOO^−^ (panel **I**), scavenging capacity of the methanolic extract (
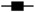
), aqueous extract (
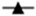
), verbenaline (3) (
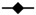
), hastatoside (4) (
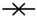
), verbascoside (5) (
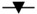
), 7OβGH (6) (
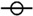
), reference compounds (
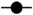
).

**Table 1 molecules-24-01970-t001:** Average recovery of the marker compounds ((3)–(6)) from *V. carolina*.

Analyte	Amount Added (μg/mL)	Amount Found (μg/mL)	Average Recovery (%)	RSD (%)
Verbenalin (3)	35	35.0943	100.26	1.98
	70	70.5581	100.79	1.89
	105	106.6606	101.58	0.60
Hastatoside (4)	20	20.216	101.08	0.48
	40	40.1022	100.25	1.15
	60	60.4068	100.68	0.60
Verbascoside (5)	45	45.0416	100.09	1.54
	90	90.7351	100.81	0.67
	135	136.0689	100.78	1.55
7OβGH, (6)	40	39.257	98.13	0.46
	80	80.4062	100.5	1.25
	120	120.7182	100.59	0.36

RSD = Relative standard deviation.

**Table 2 molecules-24-01970-t002:** Contents of verbenaline (3), hastatoside (4), verbascoside (5) and hispidulin 7OβGH (6) in six samples of *V. carolina*.

Sample	Concentration (mg/g of Plant) Mean ± RSD			
	3	4	5	6
S1	16.29 ± 0.89	11.44 ± 0.52	25.64 ± 0.67	17.87 ± 0.98
S2	17.88 ± 0.05	8.97 ± 0.16	28.82 ± 0.23	3.42 ± 0.49
S3	1.7 ± 0.00	4.39 ± 0.01	7.18 ± 0.03	6.17 ± 0.25
S4	5.83 ± 0.13	17.21 ± 0.37	31.4 ± 0.96	20.52 ± 0.90
S5	17.58 ± 0.07	14.99 ± 0.07	31.29 ± 0.09	23.71 ± 0.41
S6	0.68 ± 0.03	10.58 ± 0.10	33.78 ± 0.36	25.15 ± 0.52

Mean values (*n* = 3); RSD = Relative standard deviation.

**Table 3 molecules-24-01970-t003:** TEAC values for the DPPH, ABTS and FRAP assays.

DPPH			ABTS		FRAP	
Compound/Extract	EC_50_ (mg/mL)	TEAC [µmol trolox/mg extract]	EC_50_ (mg/mL)	TEAC [µmol trolox/mg extract]	TEAC [µmol trolox/mg extract]	AAEAC [µM AA/mg extract]
CAqE	0.081 ± 0.0001	7.50 ± 0.06	0.50 ± 0.007	7.10 ± 0.05	3.88 ± 0.06	5.49 ± 0.04
CME	0.041 ± 0.00009	14.20 ± 0.20	0.32 ± 0.01	10.10 ± 0.11	13.18 ± 0.11	11.74 ± 0.08
Ascorbic acid	0.018 ± 0.00003	23.12 ± 0.53	0.026 ± 0.0001	125.13 ± 0.90	-	-
Quercetin					89.30 ± 0.90	117.76 ± 0.61

Results are represented as the mean values ± R.S.D. *N* = 3; TEAC: Trolox equivalent antioxidant capacity; DPPH: 2,2-diphenyl-1-picryl-hydrazil-hydrate; ABTS: 2,2′-Azino-bis(3-ethylbenzothiazoline-6- sulfonic acid; FRAP: Ferric reducing ability of plasma; VCEAC: Ascorbic acid equivalent antioxidant capacity; EC_50_: 50% effective concentration; CAqE: aqueous extract; CME: methanolic extract.

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
