# Peer review of "Antimicrobial, Antioxidant Activities, and HPLC Determination of the Major Components of Verbena carolina (Verbenaceae)"

_molecules, 2019, doi:10.3390/molecules24101970_

Reviewer 1 Report

There are many papers submitted on antioxidant of some or other botanical resource. Among all those studies, this is one of the best I have seen recently. The plant is interesting and useful, it has not been much studied before, the authors use six samples to demonstrate variability (very rare to see this!), the is clear chemical analysis, antimicrobial tests, a isolation of an individual chemical of highest efficacy, and a toxicity test. The paper is well written and presented.

Author Response

Reviewer Comments:

Reviewer #1:

1. There are many papers submitted on antioxidant of some or other botanical resource. Among all those studies, this is one of the best I have seen recently. The plant is interesting and useful, it has not been much studied before, the authors use six samples to demonstrate variability (very rare to see this!), the is clear chemical analysis, antimicrobial tests, a isolation of an individual chemical of highest efficacy, and a toxicity test. The paper is well written and presented.

Authors thank Reviewer #1 for his comments.

Reviewer 2 Report

It is opinion of the reviewer that before acceptance this paper Leeds major revision. My individual comments are listed below.

The title must be changed.

L. 24 – bacteria should be listed.

L.28 – It should be “content” instead of “concentration”.

L. 29 – It should be “determined by HPLC”.

L. 30 and other places – “-O-“ must be in italic.

L. 32 – The content cannot be should be reported as %.

L. 52 – What does it mean “chemical nutrients”?
L. 52 – Digestibility of what?

L. 67 – “n-“ should be in italic.

L. 71 – It should be “The column chromatography was applied for …”. The type of column chromatography should be mentioned.

L. 82 – What does it mean “(10-5 M)”?

L. 132 – It should be “… 13.58 gallic acid equivalents/mL”.

L. 139 – It should be “A reversed …”.

L. 190 – “Abrus precatorius” should be in italic.

L. 191 – A sentence “…as ABTS and DPPH radical assays” is not clear.

L. 192 – It should be “(DPPH, ABTS, FRAP)”.

Table 2 and other places. Antioxidants are scavengers not inhibitors. Therefore, I suggest to use a term of EC50 (half maximal effective concentration) instead of IC50.

Table 2 – It should be “ABTS”. It should be “AAEAC” instead of “VCEAC”.

Table 2 – The unit if TEAC must be reported.

Figure 3 – In descriptions of the Y axis it should be “effect” instead of “capacity”.

L. 285/286 – “V. carolina” should br in italic.

L. 305 – It should be “column chromatography”.

L. 329 – It should be “EI-MS”.

L. 332 – In the text a Sephadex LH-20 column chromatography is not mentioned.

L. 333/342 should be shifted to 3.7.

L. 387 – It should be “ABTS radical cation was generated…” or “ABTS•+ was generated …”.

L. 397/398 – It should be rephrased.

L. 402 – It should be “OH”.

References – In abbreviations of the journal titles dots should be used.

Author Response

1. The title must be changed.

The title was changed as: “Antimicrobial, antioxidant activities, and HPLC determination of the major components of Verbena carolina (Verbenaceae)†”

2. L. 24 – bacteria should be listed.

Names of bacteria were listed in the text, now in line 25.

3. L.28 – It should be “content” instead of “concentration”.

The word “content” was written in the text, now in line 30.

4. L. 29 – It should be “determined by HPLC”.

The word “measured” was changed to “determined”, now in line 30.

5. L. 30 and other places – “-O-“ must be in italic.

The letter –O- was corrected as needed throughout the text.

6. L. 32 – The content cannot be should be reported as %.

This data was corrected in mg/g. Line 34.

7. L. 52 – What does it mean “chemical nutrients”?

According to investigations by Castro et al. (1991), V. carolina leaves contain among its chemical compounds: crude protein, calcium, phosphorous, iron and rivoflavin which the authors consider good nutrients for ruminants feeding. Line 54.

8. L. 52 – Digestibility of what?

Castro et al. (1991) (Inter-American Institute for Cooperation on Agriculture) tested in vitro digestibility of V. carolina leaves directed for ruminants feeding. Line 54.

9. L. 67 – “n-“ should be in italic.

Correction was already done. Line 69

10. L. 71 – It should be “The column chromatography was applied for …”. The type of column chromatography should be mentioned.

This change was done and information of the type of the column is mentioned. Line 73.

11. L. 82 – What does it mean “(10-5 M)”?

This data should not be in the text. It was removed.

L. 132 – It should be “… 13.58 gallic acid equivalents/mL”.

Correction was already done. Line 134.

L. 139 – It should be “A reversed …”.

Correction was already done. Line 140.

L. 190 – “Abrus precatorius” should be in italic.

Correction was already done in Line 191; also for Newbouldia laevis inL. 189.

L. 191 – A sentence “…as ABTS and DPPH radical assays” is not clear.

In this sentence the word “as” was changed by the most proper “in”. Line 191.

L. 192 – It should be “(DPPH, ABTS, FRAP)”.

Correction was already done. Lines 192, 193.

Table 2 and other places. Antioxidants are scavengers not inhibitors. Therefore, I suggest to use a term of EC50 (half maximal effective concentration) instead of IC50.

This term belongs to Table 3 and correction in all cases was done. In Lines 206 and 490 “inhibitory” was substituted by “effective”.

Table 2 – It should be “ABTS”. It should be “AAEAC” instead of “VCEAC”.

Corrections were done. Table 3.

Table 2 – The unit if TEAC must be reported.

The unit of TEAC is [µM trolox/mg extract] and was inserted in Table 3.

Figure 3 – In descriptions of the Y axis it should be “effect” instead of “capacity”.

Corrections in descriptions of Figure 3 were done.

L. 285/286 – “V. carolina” should br in italic.

Correction was done. Line 287-288, so as in Line 490.

L. 305 – It should be “column chromatography”.

Correction was done. Line 307.

L. 329 – It should be “EI-MS”.

Correction was done. Line 330.

L. 332 – In the text a Sephadex LH-20 column chromatography is not mentioned.

The phrase and Lipophylic Sephadex LH-20 (Sigma) was removed as this material in fact was not used.

L. 333/342 should be shifted to 3.7.

The paragraph was shifted as recommended, and now occupies L. 345/354. It was also corrected 0.45 μm in Line 342.

L. 387 – It should be “ABTS radical cation was generated…” or “ABTS•+ was generated …”.

Correction was done. Line 387.

L. 397/398 – It should be rephrased.

The last part of the paragraph was rephrased.

L. 402 – It should be “•OH”.

Correction was done. Line 402.

References – In abbreviations of the journal titles dots should be used.

All journal titles dots were inserted.

Reviewer 3 Report

The manuscript reported a method for the anaylsis of Verbena carolina L. (Verbenaceae) extracts and its effective ingredients. Where are the results on acute toxicity reported in the studies? How is dosage selected in animal studies? How is the effect of extracts compared with individual active compounds either in vitro or in vivo studies?  More updated references on Verbena carolina L. should be cited. Finally, the conclusions need to be elaborated more clearly on the potency of individual compounds. 

Author Response

Comments and Suggestions for Authors

The manuscript reported a method for the anaylsis of Verbena carolina L. (Verbenaceae) extracts and its effective ingredients. Where are the results on acute toxicity reported in the studies? How is dosage selected in animal studies? How is the effect of extracts compared with individual active compounds either in vitro or in vivo studies?  More updated references on Verbena carolina L. should be cited. Finally, the conclusions need to be elaborated more clearly on the potency of individual compounds.

1. Where are the results on acute toxicity reported in the studies?

In 2.6 of the text these results are explained as follows: For the acute toxicity test, doses of 10, 100 and 1000 mg/kg of CAqE in the first phase, and 1600, 2900 and 5000 mg/kg in the second phase were employed; an LD50 value higher than 5000 mg/kg was calculated using the geometric mean of the doses for which none of the animals died (0/3 deaths were found), and none of the organs analyzed after the experiment showed physical anomalies. Hence, CAqE of V. carolina could be regarded as non-toxic. L. 245-246.

2.  How is dosage selected in animal studies?

Dosage was selected according to protocols for testing the acute toxicity in mice as the one established by D. Lorke in 1983 (ref. 47) and the one modified by M. Déciga-Campos et al., (ref. 36).

3. How is the effect of extracts compared with individual active compounds either in vitro or in vivo studies? 

Regarding the scavenging effect, in general, individual compounds were more effective than extracts. With respect to antimicrobial activity, extracts were as actives as hispidulin against the fungi T. mentagrophites, T. rubrum and C. albicans.

4. More updated references on Verbena carolina L. should be cited.

Information showed in this work related to Verbena carolina is really updated as there is not more information related to this plant in literature, and since our work was the first to carry out the phytochemical analysis, so as the analysis of some of its biologic properties.

5. the conclusions need to be elaborated more clearly on the potency of individual compounds.

Conclusions were enriched with the proposed data. Lines 495-497

Round  2

Reviewer 2 Report

The authors corrected this paper properly taken under considerations my comments. However, in Table 3 TEAC should be expressed as “μmol trolox” istrad of “μM trolox”; M means mol/L.